



# Quantifying the nonlinear dependence of energetic electron fluxes in the Earth's radiation belts with radial diffusion drivers

Adnane Osmane[1], Mikko Savola[1], Emilia Kilpua[1], Hannu Koskinen[1], Joseph E. Borovsky[2], and
Milla Kalliokoski[1]

[1]Departmenf of Physics, University of Helsinki, Finland
[2]Space Science Institute, Colorado, USA

**Correspondence:** Adnane Osmane (adnane.osmane@helsinki.fi)

**Abstract.** In this study, we use mutual information to characterise statistical dependencies of seed and relativistic electron fluxes in the Earth's radiation belts on ultra low frequency (ULF) wave power measured on the ground and at geostationary orbit . The benefit of mutual information, in comparison to measures such as the Pearson correlation, lies in the capacity to distinguish nonlinear dependencies from linear ones. After reviewing the property of mutual information and its relationship with

the Pearson correlation for Gaussian bivariates of arbitrary correlation, we present a methodology to quantify and distinguish linear and nonlinear statistical dependencies that can be generalised to a wide range of solar wind drivers and magnetospheric responses. We present an application of the methodology by revisiting the case events studied by Rostoker et al. (1998). Our results corroborate the conclusions of Rostoker et al. (1998) that ULF wave power and relativistic electron fluxes are statistically dependent upon one another. However, we find that observed enhancements in relativistic electron fluxes correlate modestly, both linearly and nonlinearly, with the ULF power spectrum when compared with values found in previous studies (Simms

et al., 2014), and with values found between seed electrons and ULF wave power for the same case events. Our results are indicative of the importance in incorporating data analysis tools that can quantify and distinguish between linear and nonlinear interdependencies of various solar wind drivers.

## 1   Introduction

The Earth's radiation belts are nonlinearly driven and weakly collisional plasma environments in which deposited energy and momentum leads to the energisation of electrons to relativistic energies (Van Allen et al., 1958; Walt, 2005). From a fundamental physics perspective, the acceleration of charged particles to supra-thermal energies is ubiquitous to astrophysical plasma environments. As the closest astrophysical accelerator of particles to the Earth, the radiation belts are amenable to detailed *in situ* measurements of electromagnetic fields distribution functions. Their study are therefore relevant to other astrophysi-

cal environments with comparable thermodynamical properties in which particles are confined by large-scale inhomogeneous magnetic fields (Kulsrud, 2005). From an applied perspective, a wide range of satellites' orbits overlap with the Earth's radiation belts, with the undesirable consequence that the energetic particles can damage the onboard electronics and shorten the lifespan of communication systems (Baker et al., 2018). Thus, the main focus of Earth's radiation belts' studies is to quantify



the processes scaling from electron kinetic scales to planetary scales that enhance and deplete the plasma (Ukhorskiy and Sit-
nov, 2012; Thorne et al., 2013; Lejosne and Kollmann, 2020).

It has been known for several decades that the Earth's radiation belts were driven far from thermodynamical equilibrium as
a results of variable solar wind conditions (McCormac, 1965). This departure from thermodynamical equilibrium results in
kinetic distribution functions that are unstable and the production of fluctuations that can thermalise the plasma and accelerate
particles. A growing number of *in situ* measurements and observational studies in the last two decades have demonstrated that
the Earth's radiation belts' response to solar wind driving and fluctuations can also be nonlinear, and that nonlinearity ought
to be accounted for in order to improve prediction capabilities (Wing et al., 2016; Simms et al., 2018). From a theoretical
perspective, every self-consistent set of equations describing fluid and kinetic scales plasma physics are inherently nonlinear.
The departure of linearity in a dynamically evolving plasma translates into the appearance, and therefore measurements, of
non-Gaussian fluctuations (Papoulis and Pillai, 2002). Even if a nonlinear system is initialised with Gaussian fluctuations, non-
Gaussian fluctuations would eventually emerge. It is therefore not surprising that non-Gaussian fluctuations are commonly
found across a wide-range of astrophysical plasma environments (Dudok de Wit and Krasnosel'skikh, 1996; Marsch and Tu,
1997; Stepanova et al., 2003; Osman et al., 2014; Osmane et al., 2015b). Taking into account the above theoretical constraints
and observational results, one quickly recognises that in order to quantify nonlinear dependencies in the Earth's radiation belts,
one has to use measures that can be sensitive to nonlinear dependencies, and are capable to distinguish it from linear ones.

In this study, we present an application of information theory to the search of dependencies between energetic electron fluxes
measured in the Earth's radiation belts and ULF wave power measured both at geostationary orbit and on ground. Unlike more
commonly used measures like the Pearson correlation, information theoretic tools, such as mutual information, have the benefit
to distinguish nonlinear dependencies from linear ones. In order to demonstrate the value in the use of information theoretic
methods, we revisit the highly cited case studies of Rostoker et al. (1998). In their study, it was suggested that ULF pulsations
can provide energy for acceleration of electrons to relativistic energies based on visual inspection of relativistic electron fluxes
at geostationary orbit and ground ULF wave power. It should be stressed that Rostoker et al. (1998) conclusions are cautiously
stated and that a value for a correlation or any other measure is not provided. Nonetheless, it is not too uncommon to find citing
authors describing their results as compelling and evidence of strong correlation between ULF wave power and relativistic
electron fluxes. The impact of ULF fluctuations in the enhancement and loss of energetic electron fluxes also forms the basis
of radial diffusion formalisms and is, as of today, understood as one of the two dominant transport mechanisms in planetary
radiation belts (see Lejosne and Kollmann (2020) and references therein).

The application of information-theoretic measures to space plasma problems is not new but it has recently shown its utility
for a wide-range of methodologies and problems (see De Michelis et al. (2011); Wing et al. (2016); Runge et al. (2018);
Johnson et al. (2018); Osmane et al. (2019); Wing and Johnson (2019); Cameron et al. (2019) and references therein). Of
particular relevance to our study, Wing et al. (2016) presented an application of information theoretic measures to quantify



the dependence of relativistic electron fluxes measured on geostationary orbits to a wide-range of solar wind drivers. In their

study, Wing et al. (2016) demonstrate that the solar wind speed is the main driver and that the effect of the solar wind density, sometimes suggested as a dominant driver for relativistic electron fluxes (Balikhin et al.), holds 30% lesser information content and operates on a different timescale. The main departure between the work presented hereafter and the study of Wing et al. (2016) lies in our introduction of a quantity called information-adjusted correlation and the use of a dataset that has a 1 hour resolution of geostationary-measured seed and relativistic electron fluxes. The information-adjusted correlation is defined as

the correlation value that would be obtained from the mutual information under the assumption that the dependence between the two variables can be represented as a Gaussian bivariate. The choice of a Gaussian bivariate to distinguish linear and nonlinear dependences as hinted above stems from the fact that nonlinear equations produce non-Gaussian statistics even in the instance where a system is initialised with Gaussian distributed random variables (Papoulis and Pillai, 2002). We therefore present a methodology that allows us to provide clear answer the following two questions:

– (1) Are the events studied by Rostoker et al. (1998) evidence of statistical dependence between ULF wave power and electron fluxes?

   – (2) Are nonlinearities present in the instance where the dependence between ULF wave power and electron fluxes is statistically significant?

Our report is presented as follows. Section 2 presents a brief introduction to the basic tools of information theory with an

interpretation of the Shannon entropy and the mutual information. We put a particular emphasis on the application of mutual information to the case of Gaussian random variables of arbitrary correlation which serves as a benchmark for linear dependencies. In Section 3 we describe the used dataset and the associated instruments' specificities relevant to our study. In Section 4, we present our results for geostationary-measured seed and relativistic electron fluxes measured during the events presented by Rostoker et al. (1998). In Section 5, we interpret and compare our results in light of previous studies, and then conclude with

suggestions for future studies and improvement of our methodologies for instances where statistical dependencies are difficult to extract.

## 2   Methodology

In this section we present the tools we use to extract statistical dependencies of radiation belts electron fluxes on ULF wave power. The goal of this section is to clarify our methodology and provide a brief but self-contained tutorial on the Shannon

entropy and mutual information for a reader who is not familiar with information theory.

### 2.1   Mutual Information for discrete variables

It is preferable to introduce mutual information by first defining the Shannon entropy $H(X)$ for a discrete random variable $X$ (Cover, 1999). The Shannon entropy is a measure of the uncertainty contained in a random variable. In communication theory it is the number of bits on average required to describe a message $X \in \mathcal{X}$, in which $\mathcal{X}$ denotes the alphabet, or equivalently the





discrete states that can be assigned for the random variable $X$. Practically speaking, if Nadia wants to send a message to Jorge, the Shannon entropy is the average number of binary questions (e.g., yes or no) one ought to ask in order to accurately decode a message $X$ written in terms of a given alphabet $\mathcal{X}$. Mathematically, it is written in terms of the probability mass function $p(x)$ as:

$$H(X) = -\sum_{x \in \mathcal{X}} p(x) \log p(x). \tag{1}$$

The Shannon entropy is a positive definite quantity $H(X) \geq 0$ and is bounded by $H(X) \leq \log(|\mathcal{X}|)$ with equality if and only if the random variable $X$ is distributed uniformly over $\mathcal{X}$. Since the entropy is a measure of uncertainty (or equivalently knowledge), it is convenient to ask what happens to the amount of uncertainty if we are given additional information encoded in terms of $Y \in \mathcal{Y}$. In other words, do we reduce uncertainty about $X$ by knowing $Y$. In terms of probabilities for discrete events, do we gain or lose information about the likelihood of event $X$ given $Y$. Intuitively, one can assume that if $X$ and $Y$ are entirely independent, knowing one says nothing about the other. On the other hand, if $X$ and $Y$ are contingent to one another, or share a causal relationship, it can then be shown that conditioning effectively reduces entropy, and therefore uncertainty. In the instance where $X$ and $Y$ are independent, the conditional entropy $H(X|Y)$, which should be read as the entropy of $X$ given $Y$, reduces to $H(X)$. On the other hand, in the presence of some dependences, the entropy will be reduced, with $H(X|Y) < H(X)$. For two random variables $X$ and $Y$, this reduction in uncertainty is quantified by the *mutual information*:

$$
\begin{aligned}
I(X,Y) &= H(X) - H(X|Y) \\
&= \sum_{x \in \mathcal{X}} \sum_{y \in \mathcal{Y}} p(x,y) \log \left( \frac{p(x,y)}{p(x)p(y)} \right)
\end{aligned} \tag{2}
$$

The mutual information is symmetric in $X$ and $Y$ and is a measure of the dependence between two random variables. It is always nonnegative and only equal to zero if $X$ and $Y$ are independent, or equivalently if the joint distribution is the product of the marginals, i.e., $p(x,y) = p(x)p(y)$. Thus, $X$ says as much about $Y$ as $Y$ says about $X$. Additionally, since the joint Shannon entropy is $H(X,Y) = H(X) + H(Y|X)$. The proof of which is given here:

$$
\begin{aligned}
H(X,Y) &= -\sum_{x,y} p(x,y) \log p(x,y) \\
&= -\sum_{x,y} p(x,y) \log p(x)p(y|x) \\
&= -\sum_{x,y} p(x,y) \log p(x) - \sum_{x,y} p(x,y) \log p(y|x) \\
&= -\sum_{x} p(x) \log p(x) - \sum_{x,y} p(x,y) \log p(y|x) \\
&= H(X) + H(Y|X)
\end{aligned}
$$

in which we used the multiplication rule for joint probability $p(x,y) = p(x)p(y|x)$ and the requirement that $\sum_y p(x,y) = p(x)$. then $I(X,Y) = H(Y) + H(X) - H(Y,X)$ and the mutual information of one variable with itself is simply the entropy, i.e. $I(X,X) = H(X)$.





## 2.2 Mutual information for continuous variables

For a random variable $X$, if the cumulative distribution function $F(x)$ is continuous, then $X$ is said to be continuous as well. Let's denote the probability distribution function $f(x) = dF(x)/dx$. The *differential entropy* of a continuous random variable $X$ is defined as:

$$h(X) = -\int_{S \in \mathbb{R}} f(x) \log f(x) \, dx \tag{3}$$

where $S$ is the support set where $f(x) > 0$. Differential entropy $h(X)$, as in the discrete case with the Shannon entropy $H$, is also a measure of the uncertainty for a random variable $X$. However, unlike in the discrete case, the differential entropy can be negative. Consider for instance a random variable distributed uniformly from $0$ to $L$, so that its density is $f(X) = 1/L$. Then its differential entropy is:

$$\begin{aligned} h(X) &= -\int_0^L \frac{1}{L} \log \frac{1}{L} \, dx \\ &= \log(L) \end{aligned} \tag{4}$$

Thus, for $L < 1$, $\log L < 0$ and the differential entropy is negative. The mutual information $I(X;Y)$ can be extended to continuous variables as:

$$\begin{aligned} I(x;y) &= \int f(x,y) \log \frac{f(x,y)}{f(x)f(y)} \, dx \, dy \\ &= -\int f(x,y) \log f(x) \, dx \, dy - \int f(x,y) \log f(y) \, dx \, dy \\ &\quad + \int f(x,y) \log f(x,y) \, dx \, dy \\ &= h(x) + h(y) - h(x,y) \end{aligned} \tag{5}$$

## 2.3 Relationship between the Pearson correlation and mutual information for Gaussian bivariates

Even though probability distribution functions of electromagnetic fields and particle velocity distributions in space plasmas often depart from Gaussianity, it is useful to refer to the Gaussian bivariate case to develop an appreciation of mutual information for linear systems and as a benchmark to test numerical estimates. Conveniently, there is an exact analytical relationship between the Pearson correlation and mutual information of a Gaussian bivariate. As a particular case, we consider a bivariate $\boldsymbol{X} = (X, Y)^T$ with mean vector

$$\boldsymbol{\mu} = \begin{pmatrix} \mu_x \\ \mu_y \end{pmatrix} \tag{6}$$

and covariance matrix given by

$$\boldsymbol{C} = \begin{pmatrix} \sigma_x^2 & \sigma_x \sigma_y \rho \\ \sigma_x \sigma_y \rho & \sigma_y^2 \end{pmatrix}$$





for means $E[X] = \mu_x$, $E[Y] = \mu_y$, variances $\sigma_x^2 = E[X^2] - \mu_x^2$, $\sigma_y^2 = E[Y^2] - \mu_y^2$ and correlation coefficient $\rho$ defined as:

$$\rho = \frac{E[XY] - \mu_x\mu_y}{\sigma_x\sigma_y}. \tag{7}$$

The probability density function of the $X - Y$ bivariate is:

$$
\begin{aligned}
f(x,y) &= \frac{1}{2\pi|\boldsymbol{C}|^{1/2}} \exp\left(-\frac{1}{2}\boldsymbol{X}^T\boldsymbol{C}^{-1}\boldsymbol{X}\right) \\
&= \frac{1}{2\pi\sigma_x\sigma_y\sqrt{1-\rho^2}} \exp\left[-\frac{1}{2(1-\rho^2)}\left(\frac{(x-\mu_x)^2}{\sigma_x^2}\right)\right] \\
&\quad \times \exp\left[-\frac{1}{2(1-\rho^2)}\left(\frac{(y-\mu_y)^2}{\sigma_y^2} - \frac{2\rho(x-\mu_x)(y-\mu_y)}{\sigma_x\sigma_y}\right)\right]
\end{aligned}
\tag{8}
$$

For the sake of simplicity we focus on the case where $\mu_x = \mu_y = 0$ and $\sigma_x = \sigma_y = \sigma$, in which case the joint bivariate distribution takes the form:

$$f(x,y) = \frac{(1-\rho^2)^{-1/2}}{2\pi\sigma^2} \exp\left(-\frac{x^2 + y^2 - 2\rho xy}{2(1-\rho^2)\sigma^2}\right) \tag{9}$$

and the marginals $f(x_i) = (2\pi\sigma^2)^{-1/2}\exp\left(-x_i^2/2\sigma^2\right)$ for $x_i = (x,y)$. Using equation (5) we can compute the mutual information between $X$ and $Y$. For $h(x_i)$ we find:

$$
\begin{aligned}
h(x_i) &= -\int f(x_i)\log f(x_i)\, dx_i \\
&= \frac{1}{2}\int f(x_i)\log 2\pi\sigma^2\, dx_i + \frac{1}{2\sigma^2}\int f(x_i)x_i^2\, dx_i \\
&= \frac{1}{2}\log 2\pi\sigma^2 + \frac{1}{2} \\
&= \frac{1}{2}\log 4\pi\sigma^2
\end{aligned}
\tag{10}
$$

in which the logarithm is in base 2. And now for the joint differential entropy of a Gaussian bivariate:

$$
\begin{aligned}
h(x,y) &= -\int f(x,y)\log f(x,y)\, dx\, dy \\
&= -\log\frac{(1-\rho^2)^{-1/2}}{2\pi\sigma^2}\int f(x,y)\, dx\, dy \\
&\quad - \int\left(\frac{x^2 + y^2 - 2\rho xy}{2(1-\rho^2)\sigma^2}\right)f(x,y)\, dx\, dy \\
&= \frac{1}{2}\log(1-\rho^2) + \log 4\pi\sigma^2.
\end{aligned}
\tag{11}
$$

Therefore, the mutual information of a Gaussian bivariate is a nonlinear function of the correlation $\rho$:

$$
\begin{aligned}
I(x,y) &= h(x) + h(y) - h(x,y) \\
&= -\frac{1}{2}\log(1-\rho^2).
\end{aligned}
\tag{12}
$$

Since the mutual information is a measure of how much we know from $X$ given $Y$, and vice-versa, the nonlinearity between mutual information and the correlation is an indication that the latter should not be interpreted linearly. Indeed, the difference





in information between random variables of 0.75 and 0.5 correlation is not of order 50% ($0.75/0.5 - 1 = .5$) but rather $187\%$. Thus, two random variables with Pearson correlation of 0.75 carry a much larger amount of information upon one another than one with correlation of 0.5. An additional constraint with the Pearson correlation resides with fat-tailed random variables. For Gaussian bivariates, independence is synonymous with uncorrelated. However, for fat-tailed random variables, as commonly measured in space and astrophysical plasmas, strongly dependent random variables can have zero correlation (Taleb, 2020). Unlike the Pearson correlation, mutual information is able to quantify the dependence of random variables in the absence of correlation. As a simple example the reader can test for itself, we here assume two random variables $X$ and $Z$. $X \sim N(0,1)$ is Gaussian random variable with zero mean and a standard deviation of 1. $Z$ is the square of $X$, i.e. $Z = X^2$. Thus the relationship between $Z$ and $X$ is nonlinear and there is no doubt that $Z$ and $X$ are statistically dependent with one another. However, computing the Pearson correlation is inconclusive as it gives a value of zero, whereas mutual information computed with the code describe below indicates a large statistical dependence with a value of 1.42.

### 2.4 Numerical computation of mutual information

The procedure we follow to compute the mutual information for two time series consist in binning the data according to the Freedman-Diaconis rule (Freedman and Diaconis, 1981). Thus, even though the electron fluxes and ULF wave power are continuous variables, our procedure has the consequence to discretise the variables. This discretisation leads to biases in the estimation of mutual information that are dependent on the number of measurement points $N$ and statistical dependence of the two variables. For instance, two Gaussian random variables with a high correlation would require less measurement points $N$ to estimate the mutual information than two Gaussian random variables with a low correlation or two fat-tailed random variables with some arbitrary correlation.

The first test for a mutual information estimator can be performed for a Gaussian bivariate of arbitrary correlation $\rho$. In Figure 1 we plotted the numerical estimate and analytical solution for $10^6$ points extracted from Gaussian bivariates with correlations ranging between $-1$ and 1. We note that our estimator does well for low correlation values though it gains a discrepancy of order $10\%$ for correlation absolute values greater than 0.5. In order to estimate the error introduced by discretization we apply a shuffle test to the two time series and compute the average value of mutual information and its standard deviation for one hundred shuffles. We find that the error computed with the shuffling procedure is Gaussian distributed and we interpret the average mutual information obtained from shuffling as the zero baseline level. This baseline for each events studies is plotted as an orange bold line in Figures 4-11 for panels (a) and (c). The shaded orange area represents the 3 standard deviation range from the mean. Estimates of mutual information for electron fluxes and ULF wave power above the shaded area are therefore interpreted as significant with $\pm 3$ standard deviation. More sophisticated methods to compute mutual information through non-parametric methodologies are possible, but for our dataset, the statistical dependence between variables and the number of points are sufficient for us to answer the questions stated in the introduction.





## 3 Dataset

The data used in this study corresponds to the two events analysed by Rostoker et al. (1998). The first period extends from March second to May 31st in 1994 (91 days in total) and the second one spans from November first to 26th in 1993. During the first period a big geomagnetic storm occurred on April 17th with minimum Dst of -201 nT, and the period featured also several moderate and intense storms. During the second period an intense storm peaked on November 4th with minimum Ds -119 nT. Another significant storm during this latter period was a moderate storm on November 18th with Dst minimum -82 nT. Both periods were thus geomagnetically active. Our choice to revisit the work of Rostoker et al. (1998) through mutual information stems from the fact that such methodology has not been used before, and that their study, highly cited in the literature as evidence that radial diffusion is a leading mechanism for the energisation of relativistic electrons, can serve as a benchmark for more involved methodologies. Additionally we have access to a comparable dataset with better resolution (1 hour resolution instead of 1 day), so we can not only revisit the results of Rostoker et al. (1998) with information theory but find a more accurate time lag for the electron's response to ULF wave power. In Rostoker et al. (1998) the Pc5 ULF measurements were from the Gillam measurement station of the Canadian Auroral Network for the OPEN Program Unified Study Project (CANOPUS) and the electron fluxes ($> 2$ MeV) were from Geostationary Operational Environmental Satellite 7 (GOES 7). The GOES data is the daily average flux and the ULF data is the average over a six hour period from dawn to noon.

### 3.1 ULF power spectrum

The ULF data used in this analysis was from National Aeronautics and Space Administration's (NASA) Virtual Radiation Belt Observatory (ViRBO) and the used ULF indices Sgr and Sgeo, both describing ULF spectral power from which noise has been removed, are derived in Kozyreva et al. (2007) The ULF data is for Pc5 frequency range of 2 - 10 mHz. The ULF indices used in this work are the logarithm in base 10 of the signal spectral power. The signal spectral power is the integral over the power spectral density above the noise level (Kozyreva et al., 2007). The index values of signal spectral power are 1 hour averages from measurements done in 1 minute resolution by a global network of measuring stations. The measurements of each station are averaged separately, and the index value is the maximum of those hourly averages. The *in situ* geosynchronous index $S_{geo}$ has been calculated from the measurements of GOES spacecraft and the ground ULF index $S_{gr}$ is based on measurements from stations on the Northern hemisphere. The ULF measurements from ground stations for any hour of universal time have been done in the MLT sector from 3 to 18 hours and between the CGM latitudes $60^o$ and $70^o$. For the ground index, omitting the stations outside the MLT sector from 5 to 15 hours has little effect on the measurement results, since the cross-correlation between the ULF measurements in the MLT interval of 3 to 18 hours and those that span the MLT interval 0 - 24 hours is about 0.95 (Kozyreva et al., 2007). It is interesting to compare both ground and geostationary ULF activity since toroidal ULF waves with small azimuthal mode number $m$ waves can transmit to the ground, whereas poloidal ULF waves with high azimuthal mode number $m$ are confined to the inner magnetosphere. Azimuthal mode number affects the electron energies that can resonate with these waves and a discrepancy in correlational measures for ground and geostationary ULF measurements can be of indicative of certain wave mode dominance.





### 3.2 Seed and relativistic electron fluxes indices

In order to quantify the electron fluxes we use the indices $F_{e1.2}$ and $F_{e130}$ described in Borovsky and Yakymenko (2017) for electrons with energies near 1.2 MeV and 130 keV, respectively. The indices are computed as the base 10 logarithm of the maximum geostationary- measured electron fluxes by any of the SOPA instrument for a given energy channel at the outer radiation belts. For every hour of universal time the maximum of 6 minute median values over all satellites is recorded as the flux value during that hour. The median values for each satellite are calculated from measurements done at a 10 second sampling rate[1]. $Fe_{130}$ is a measure of the intensity of substorm-injected electrons in the dipolar magnetosphere: $Fe_{130}$ rises rapidly at the onset of a magnetospheric substorm and subsequently decays over the timescale of a few hours. $F_{e1.2}$ is a measure of the intensity of the outer electron radiation belt: $F_{e1.2}$ grows slowly during very active times and decays over the timescale of a few days during quiet times. $F_{e1.2}$ can also exhibit sudden dropouts at the onsets of geomagnetic storms.

## 4 Results

Figures (2) and (3) show the 24 hour average of relativistic electron flux indices and ULF power indices as a function of time for the two events studied by Rostoker et al. (1998). In each figure the panel on the left has the geosynchronous ULF index plotted, whereas the panel on the right has the ground ULF index plotted. We remind the reader that our data sets have different time resolutions from those used by Rostoker et al. (1998) with 24 hours resolution, whereas we use 1 hour resolution and 24 hours moving averages. However, the visual comparison of Figures (2) and (3) to Figures (1) and (2) in Rostoker et al. (1998) show that they are very similar [2]. In the following we will look at each event separately and compare the values obtained for the mutual information and the Pearson correlation.

### 4.1 Event 1 for the electron index $F_{e1.2}$

Figure (4) shows the mutual information and correlation of the relativistic electron flux index $F_{e1.2}$ with ULF wave power as a function of time lag for the Event 1 from March 1 to May 31st in 1994 of Rostoker et al. (1998). The increment in time lag is of 1 hour. Positive time lag indicate that changes in ULF wave power precedes those in electron flux and positive vice-versa. The panels (a) and (b) in each figure show the dependence with ULF ground index $S_{gr}$, whereas panels (c) and (d) are for the dependence with ULF geostationary index $S_{geo}$. The orange line in the panels with mutual information represents the zero value on the basis of the shuffling procedure described in the methodology section. The shaded area overlapping the zero curve for mutual information represents the six standard deviation spread. Thus a value above the shaded area represents a measurement of mutual information that has at least six sigma significance. We note that the peaks in mutual information and

---

[1]The fluxes have been derived in Cayton and Belian (2007) by converting them from count rates. The electron counts contain also incident protons, alpha particles, and gamma-rays, which have been treated as additional electrons instead of being removed from the raw data. Changes in processing the measurement data over the years may also have caused systematic errors in the measurement data, but maybe only a few percent of the data records are contaminated (Cayton and Belian, 2007)

[2]Reducing our resolution to 24 hours for a strict comparison with (Rostoker et al., 1998) is not useful because the values of mutual information and correlation are low, and reducing the number of points would bring both measures to the noise level.





Pearson correlation occur between 48 and 50 hours time lag and have maximum values of $I_{max} \simeq 0.5$ and $\rho_{max} \simeq 0.55 - 0.6$. Also, the mutual information and correlation of electron fluxes with geostationary ULF power $S_{geo}$ shows a prominent 24 hour modulation. As is typical for an index that measures magnetospheric quantities, $F_{e1.2}$, $F_{e130}$, $S_{gr}$, and $S_{geo}$ have 24-hr periodicities in them caused by dipole wobble, longitudinal station coverage, etc. These 24-hr periodicities show up as 24-hr peaks in their autocorrelation functions (see Fig. 2a of Borovsky and Yakymenko (2017) for $F_{e1.2}$ and $F_{e130}$ and see Fig. 4a and 4b of Borovsky and Denton (2014) for $S_{gr}$ and $S_{geo}$). These 24-hr periodicities will also show up in the cross-correlations between magnetospheric variables.

Figure (5) looks at the same dependence as in Figure (4) but for a time moving 24 hour average of the indices. Using a time moving average introduces statistical dependence between points less than 12 hour lag apart, but is useful to denote long-term trends. The mutual information and correlation in Figure (5) have the same peaks and shape as in Figure (4) for the one hour resolution, but because of the averaging the modulation present in the high resolution data is lost.

### 4.2 Event 2 for the electron index $F_{e1.2}$

Figure (6) shows the mutual information and correlation of the relativistic electron flux index with ULF wave power as a function of time lag for the Event 2 from November 2nd to November 26th in 1993 of Rostoker et al. (1998). Similarly to Figure (4), indices are plotted with a 1 hour time lag increment. The panels (a) and (b) in each figure show the dependence with ULF ground index $S_{gr}$, whereas panels (c) and (d) are for the dependence with ULF geostationary index $S_{geo}$. We note that for Event 2 local peaks occurs for 24-48 hours lag time, but both the mutual information and Pearson correlation, for comparable resolution, are significantly weaker than for Event 1 with $I_{max} < 0.4$ $\rho_{max} < 0.5$. Unlike for Event 1, Event 2 shows different dependence on the time lag between the mutual information and correlation. This discrepancy between the two measures could be indicative of time dependent nonlinearity of relativistic electron fluxes with ULF wave power.

Figure (7) looks at the same dependence as in Figure (6) but for a time moving 24 hour average of the indices. A 24 hour running average removes the 24 hour periodicity in the indices and hence removes the 24 hour peaks in the cross-correlations. We note that the value of the mutual information is once again significantly enhanced since the averaging introduces statistical dependencies between two points less than 12 hours apart, but we also notice that there is a different dependence than for the Pearson correlation. These differences between the two measures and their potential origin in nonlinear phenomena are discussed in the Discussion section.

### 4.3 Event 1 for the electron index $F_{e130}$

Figure (8) shows the mutual information and correlation of 130 keV electron flux index $F_{e130}$ with ULF wave power as a function of time lag for the Event 1 of Rostoker et al. (1998). The indices are once more plotted with a 1 hour resolution and time lag increment. The panels (a) and (b) in each figure show the dependence with ULF ground index $S_{gr}$, whereas panels (c) and (d) are for the dependence with ULF geostationary index $S_{geo}$. We note that the time lag dependence of mutual information





and correlation are comparable and that the peak in both occurs for a lag of $\tau = 0$. The peak in the mutual information between $S_{gr}$ and $F_{e130}$ is $I_{\max} \simeq 0.68$, which is significantly greater than the mutual information between $S_{gr}$ and $F_{e1.2}$. On the other hand, the peak in the mutual information between $S_{geo}$ and $F_{e130}$ is $I_{\max} \simeq 0.4$, which is comparable to the peak value we found for the mutual information between $S_{geo}$ and $F_{e1.2}$. As observed in Figure (4) we also note a modulation in the mutual information and correlation of electron fluxes with geostationary ULF power $S_{geo}$ not present in the dependence with the

ground power index $S_{gr}$. Figure (9) shows the same dependence as in Figure (8) but for a time moving 24 hour average of the indices.

### 4.4  Event 2 for the electron index $F_{e130}$

Figure (10) has the same description as Figure (8) but the mutual information and Pearson correlation are computed for Event 2 of Rostoker et al. (1998). Similarly to Event 1, Event 2 shows that the time lag dependence of mutual information and

correlation is comparable and that the peak in both occurs around a lag of $\tau = 0$ and values of $I_{\max} \simeq 0.6 - 0.68$. Figure (11) looks at the same dependence as in Figure (10) but for a time moving 24 hour average of the indices. A comparison of Event 1 and Event 2 show a similar time response and dependence of 130 keV electron flux index $F_{e130}$ with ULF wave power.

## 5  Discussion

We are now ready to answer the two questions stated in the introduction: (1) Are the events studied by Rostoker et al. (1998)

examples of strong ULF wave power and energetic electron dependence? (2) Is the statistical dependence between ULF wave power and electron fluxes nonlinear? In order to answer these two questions we have tabulated the values of the maximum Pearson correlation and maximum mutual information for all events in Table 1. The columns denote, from the left to the right, the event year, the flux index, the ULF index, the maximum Pearson correlation, the maximum mutual information, the information-adjusted correlation, and the lag for the maximum mutual information, respectively. The information-adjusted

correlation is defined as the correlation value that would be obtained from the mutual information under the assumption that the dependence between the two variables can be represented as a Gaussian bivariate (cf. Equation 12). The choice of a Gaussian bivariate to distinguish linear and nonlinear dependences stems from the fact that nonlinear equations produce non-Gaussian statistics even in the instance where a system is initialised with Gaussian distributed random variables (Papoulis and Pillai, 2002). Mathematically, the information adjusted correlation can be defined by applying the inverse of Equation (12):

$$\rho_{adj} = \text{sign}(\rho)\sqrt{1 - 2^{-2I}} \qquad (13)$$

The information-adjusted correlation $\rho_{adj}$ allows us to determine whether the Pearson correlation has underestimated the dependence between the random variables due to the presence of nonlinearity. The instance in which the adjusted correlation is statistically comparable to the Pearson correlation denotes that a linear dependence between the fluxes and ULF power dominates and that nonlinear dependencies are either too weak or non-existent. In the opposite case, an adjusted correlation larger



than the Pearson correlation indicates that nonlinear dependencies between fluxes and ULF power are statistically significant.

Are the events evidence of strong ULF wave power and energetic electron dependence? For the two events studied, the Pearson correlation and the mutual information are both statistically significant, and well above the noise level. However, the maximum correlation values for relativistic electrons range between 0.41 and 0.59, and the maximum mutual information values range

between 0.36 and 0.49. For comparisons, Simms et al. (2014) analyses from 219 storms find values of correlation of 0.65 for ground ULF and 0.50 for GOES ULF with relativistic electron fluxes. Thus, the values for correlation of the 1993 event are showing weaker linear and nonlinear statistical dependence whereas the 1994 event has correlation values on par with events found over 11 years of data (Simms et al., 2014). The methodology of Simms et al. (2014) separates variables in terms of storm phases and defines a predictor variable, e.g., ULF wave power, as an average over an appropriate time period for a given storm

phase. Since we are studying case events the statistical methodology of Simms et al. (2014) can not be explicitly reproduced, but we find comparisons with our results useful in that it gives us a point of reference to judge the strength of the correlation values we found. For another comparison, for all times during the years 1995-2006 (Borovsky, 2017) found a Pearson correlation coefficient of 0.34 between $F_{e1.2}$ and $S_{gr}$, whereas they found a higher correlation coefficient of 0.54 between $F_{e1.2}$ and the 123-hr time integral of $S_{gr}$. Similarly it was found that the correlation coefficient between $F_{e1.2}$ and $S_{geo}$ was 0.21

whereas the correlation coefficient was 0.25 between $F_{e1.2}$ and the 138-hr time integral of $S_{geo}$. For all times during the years 1995-2004 Borovsky and Denton (2014) explored correlation coefficients between $S_{gr}$ and $S_{geo}$ and a relativistic-electron flux $F$ that was calculated differently from $F_{e1.2}$. They found correlations between $F$ and $S_{gr}$ of 0.36 (with a time lag of 56 hr) and between F and the 126-hr time integral of $S_{gr}$ of 0.55. Likewise they found correlations between F and $S_{geo}$ of 0.28 (with a time lag of 71 hr) and between F and the 156-hr time integral of $S_{gr}$ of 0.32. Our results demonstrate that even though the

events appear, at least visually, to show strong correlation between ULF waves and relativistic electron fluxes, quantitatively the dependence is significant but modest.

Comparing between seed and relativistic electrons, the statistical dependence with ULF wave power of the 130 keV flux is significantly larger than for relativistic fluxes and ranges between 0.54 and 0.68 for the maximum Pearson correlation and

0.44 and 0.67 for the maximum mutual information. We also note that the time-lag for the maximum values is comparable whether one uses the mutual information or the Pearson correlation. 130 keV fluxes have a maximum dependence with time lags of less than a day whereas the relativistic electrons see a maximum for time lags considerably longer between 42 and 67 hours. Moreover, the ground ULF wave power gives a larger dependence than geostationary measured ULF wave power for the 1994 event. For the 1993 event the statistical dependence is the same whether one uses ground or geostationary indices.

The ground ULF index spans local daylight hours between 0500 and 1500, whereas the GOES ULF covers the full 24 hours period. This local time difference between ground and geostationary sampling of wave power makes the latter more susceptible to be influenced by substorm activity and the former by viscous processes and pressure pulses on the dayside magnetosphere during moderate geomagnetic activity (Borovsky and Funsten, 2003; Osmane et al., 2015a). However, and as pointed out by Simms et al. (2014), the most notable difference between ground and GOES data is that the ground magnetometers are better





positioned to catch ULF wave activity that would result in radial diffusion transport (Lejosne and Kollmann, 2020).

To address the second question, we compare the values of the information adjusted correlation with the Pearson correlation. We note that the adjusted correlation is significantly larger than the Pearson correlation for all instances. In other words, though constrained to two case studies, our results demonstrate the presence of nonlinear statistical dependencies between energetic

electron fluxes and ULF wave power. By using information theory we make no assumptions about the functional form of the nonlinear dependence between the variables, but we can nonetheless state that nonlinearities have to be accounted for. Our results are consistent with the study of Simms et al. (2018) in which they built regression models that assumed a quadratic dependence in the ULF wave power with a one day lag. Their results indicate that the response of relativistic electron fluxes can be a combination of linear and nonlinear dependence and that incorporating a quadratic term might provide better predictions.

Based on the values for the information adjusted correlation, the Pearson correlation might be missing between 20% and 30% of the statistical dependencies between ULF wave power and relativistic electron fluxes.

## 6    Conclusions

The Earth's inner magnetosphere is a nonlinearly driven plasma environment in which electrons can be collectively energised to relativistic energies by ULF fluctuations (Lejosne and Kollmann, 2020). The emergence of nonlinear processes translates

into non-Gaussian fluctuations in the electromagnetic fields and particle distribution functions. Thus, in order to quantify the processes at play to model the Earth's radiation belts accurately, one needs to determine whether nonlinear statistical dependencies between drivers, such as the solar wind speed and the ULF wave power, and quantities in which energy and momentum is deposited , such as electron fluxes, have to be accounted for.

In this study, we described the use of mutual information to characterise statistical dependencies of relativistic electron fluxes on ULF wave power. The benefit of mutual information, in comparison to the Pearson correlation, lies in the capacity to distinguish nonlinear dependencies from linear ones. In order to test our methodology, we revisited the case study of Rostoker et al. (1998) in which two events were shown, from a visual perspective, to indicate strong correlation between the rise of relativistic electron fluxes and ULF wave power. Our application of mutual information to the events presented by Rostoker

et al. (1998) indicate that relativistic electron fluxes are linearly and nonlinearly dependent on ULF wave power. However, the values that we found for both the Pearson correlation and mutual information are modest when compared to previous statistical results (Simms et al., 2014) and consistently smaller than the one found between seed electrons and ULF wave power. Our results do not indicate a necessary causal physical relationship between seed electrons and ULF wave power, but it does point out to the necessity of not over-interpreting correlational measures, whether linear or nonlinear. The modest dependence of

relativistic electron fluxes with ULF wave power could also originate in a shared dependence on solar wind drivers such as the solar wind speed. Our results are therefore indicative of the need to incorporate data analysis tools that can distinguish between interdependencies of various solar wind drivers. In the framework of information theory, conditional mutual information is



specifically built for that purpose and has been successfully used to resolve a long lasting question about the relative role of solar wind speed and density in driving relativistic electron fluxes (Wing et al., 2016). In future studies, we will also apply a

comparable methodology presented in (Simms et al., 2014) to seek dependencies of relativistic electron fluxes on solar wind drivers for given storm phases and build non-parametric estimators for the probability density of random variables that do not require binning (Kraskov et al., 2004).

*Data availability.* Data for the relativistic electron fluxes can be requested to Joe Borovsky. Data for ULF wave power can be found on the Augsburg website.

*Author contributions.* AO designed the study, wrote most of the article and participated in the interpretation of results, MS wrote the code, the methodology section, performed the data analysis and participated in the interpretation of results. EK, HK, JB and MK participated in the interpretation of results and literature review.

*Acknowledgements.* AO is grateful for enlightening conversations with Simon Wing, Jay Johnson and Solene Lejosne on the topics of information theory and radial diffusion transport of radiation belts.





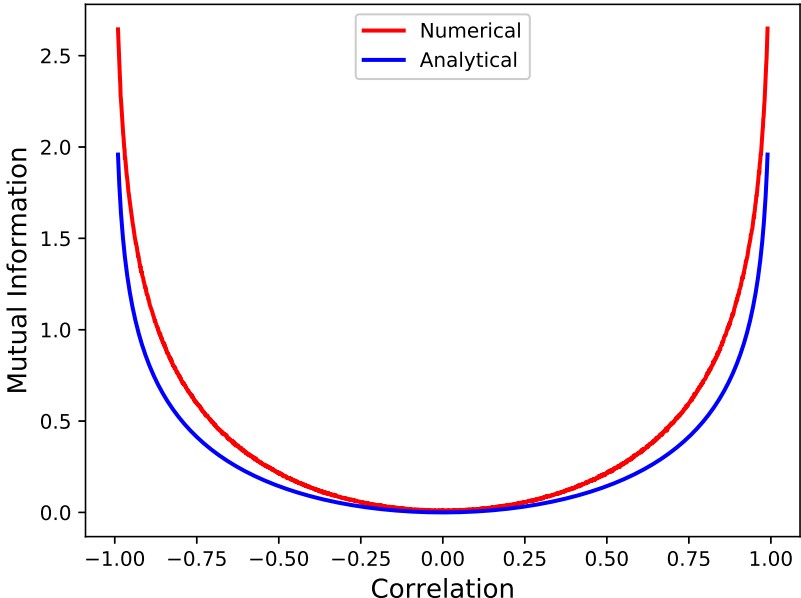

**Figure 1.** Mutual information estimator for bivariate Gaussian random variables with $\sigma_x = \sigma_y = 1$ and zero mean as a function of the correlation $\rho \in [-1, 1]$. The numerical value is traced in red, and the analytical in blue. The estimator is computed for $N = 10^5$ points and $46 \ (\simeq N^{1/3})$ bins according to the Freedman-Diaconis rule.

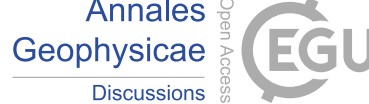

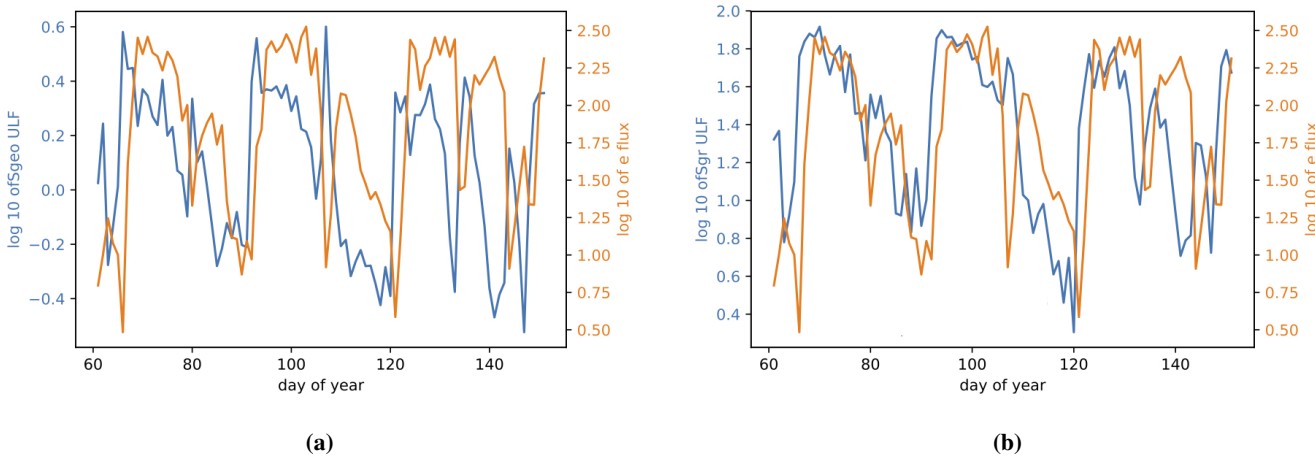

**Figure 2. (a)** 24 hour average of relativistic electron flux index $F_{e1.2}$ and geosynchronous ULF index $S_{geo}$ for the 1994 event from the Rostoker et al. (1998) study. **(b)** Same as for panel (b) but with ground ULF index $S_{gr}$ for the 1994 event. We note that visually the ground index $S_{gr}$ follows more closely the relativistic electron flux index for the duration of the event.

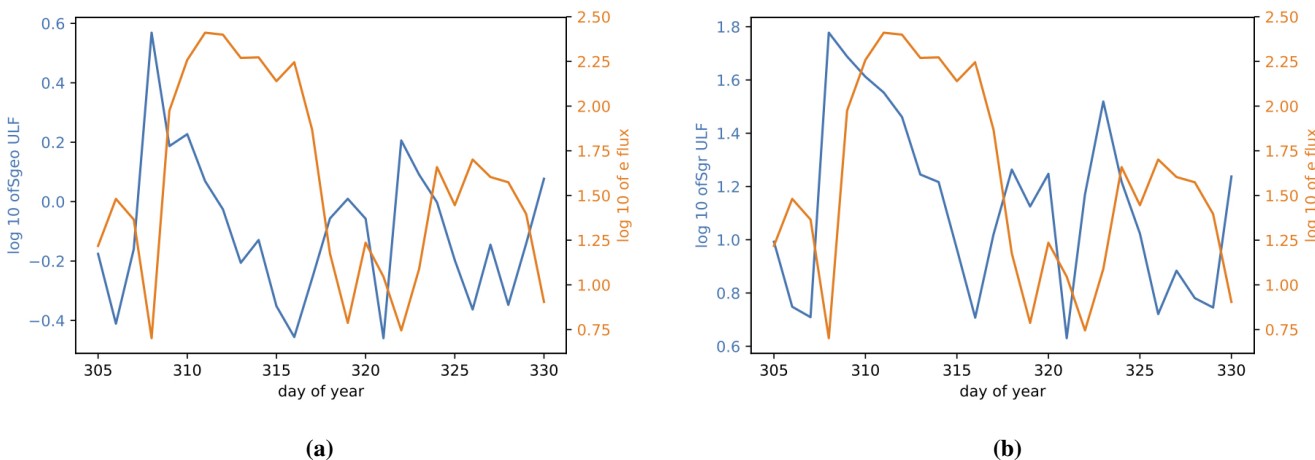

**Figure 3. (a)** 24 hour average of relativistic electron flux index $F_{e1.2}$ and geosynchronous ULF index $S_{geo}$ for the 1993 event from the Rostoker et al. (1998) study. **(b)** Same as for panel (b) but with ground ULF index $S_{gr}$ for the 1993 event.



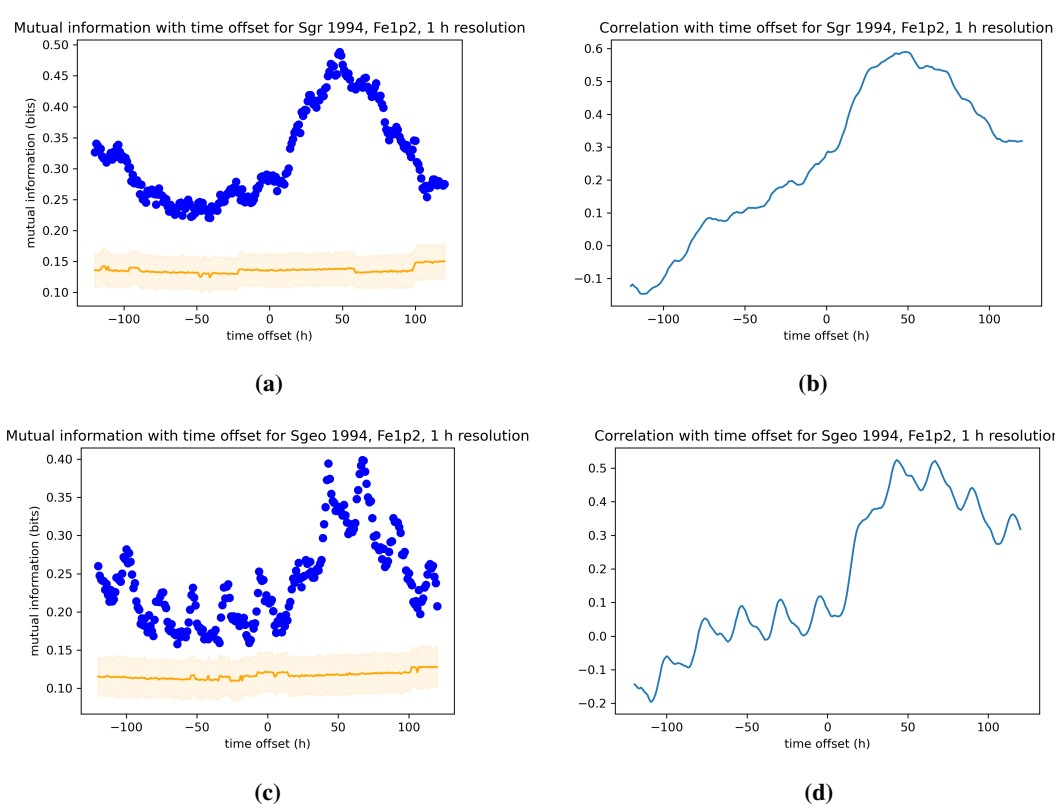

**Figure 4.** Dependence of the mutual information and Pearson correlation for ground ((**a**) and (**b**) ) and geosynchronous ( (**c**) and (**d**) )ULF power with the relativistic electron index $F_{e1.2}$ for the 1994 Event. The data for both indices is processed with 1 hour resolution. See subsection 4.1 for a description of the figures.



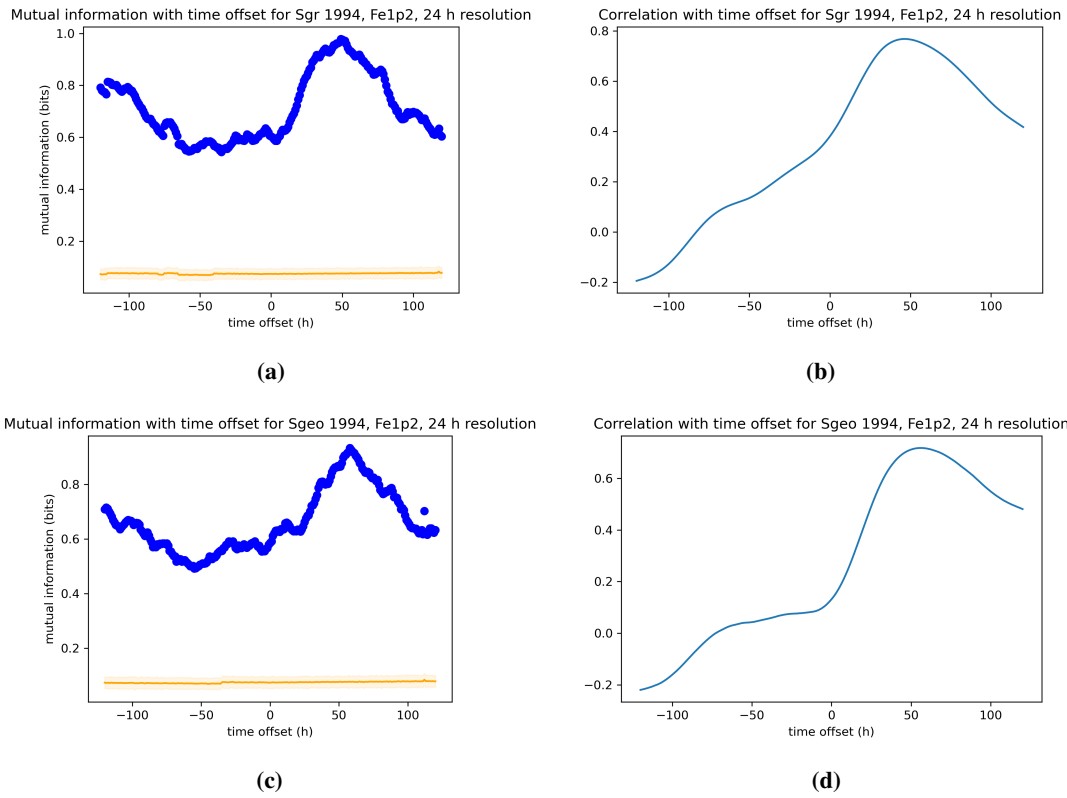

**Figure 5.** Same as in Figure 4 but with a 24 hour moving average of each indices.

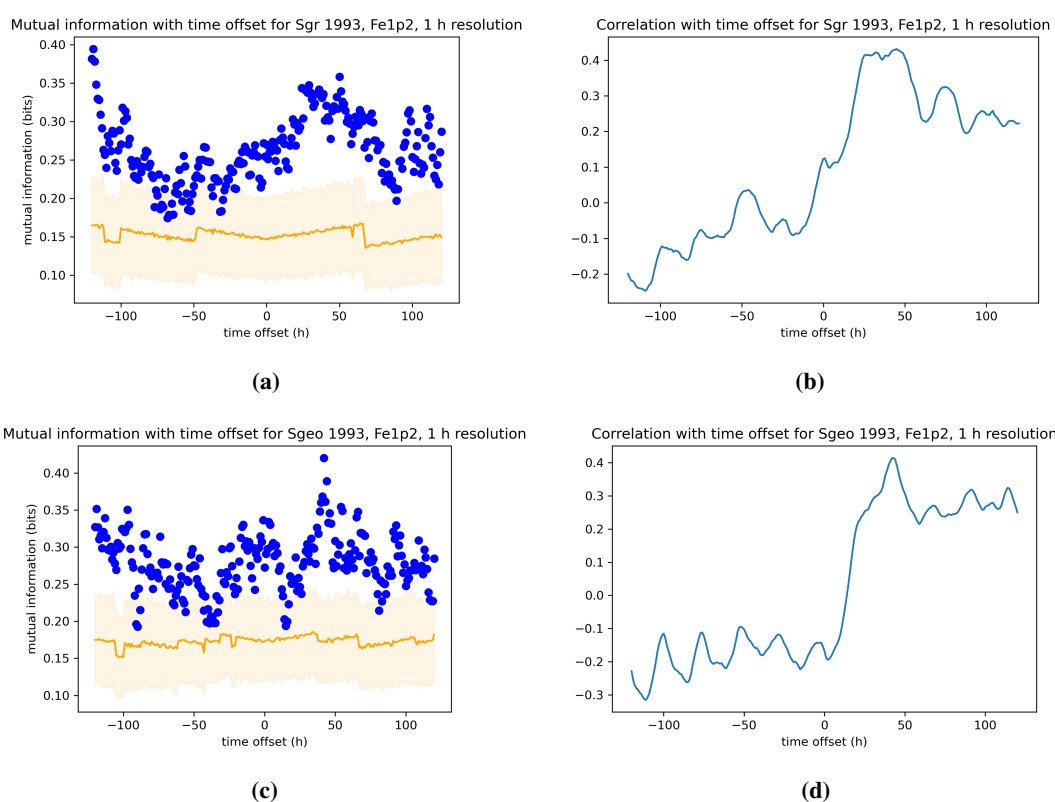

**Figure 6.** Same as in Figure 4 but for the 1993 event of Rostoker et al. (1998).

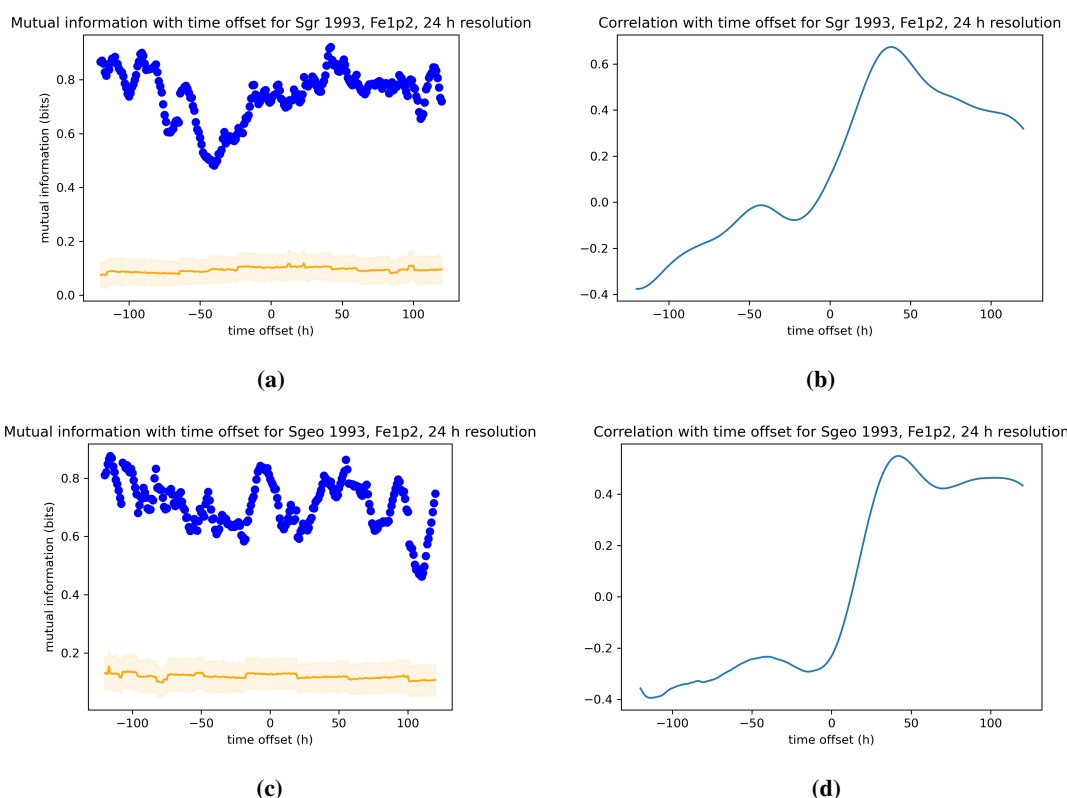

**Figure 7.** Same as in Figure 5 but for the 1993 event of Rostoker et al. (1998).

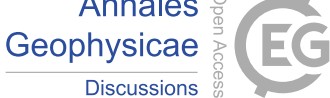

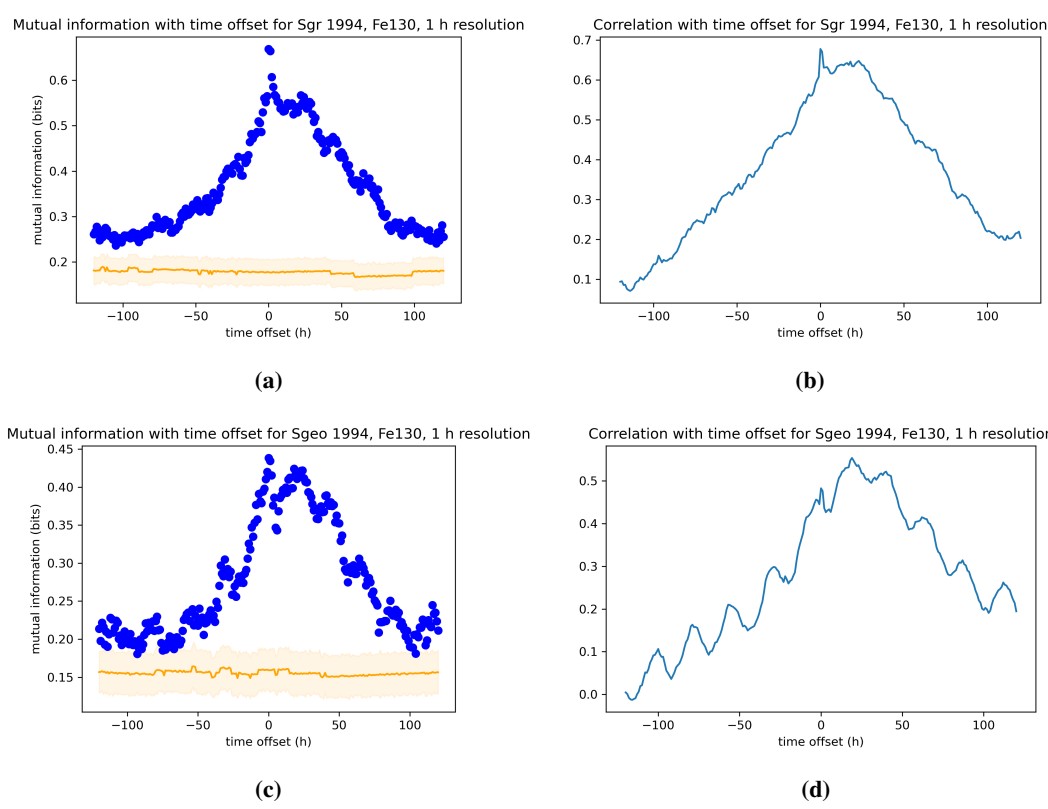

**Figure 8.** Dependence of the mutual information and Pearson correlation for ground ((**a**) and (**b**) ) and geosynchronous ( (**c**) and (**d**) )ULF power with the 130 keV electron index $F_{e130}$ for the 1994 Event. The data for both indices is processed with 1 hour resolution.

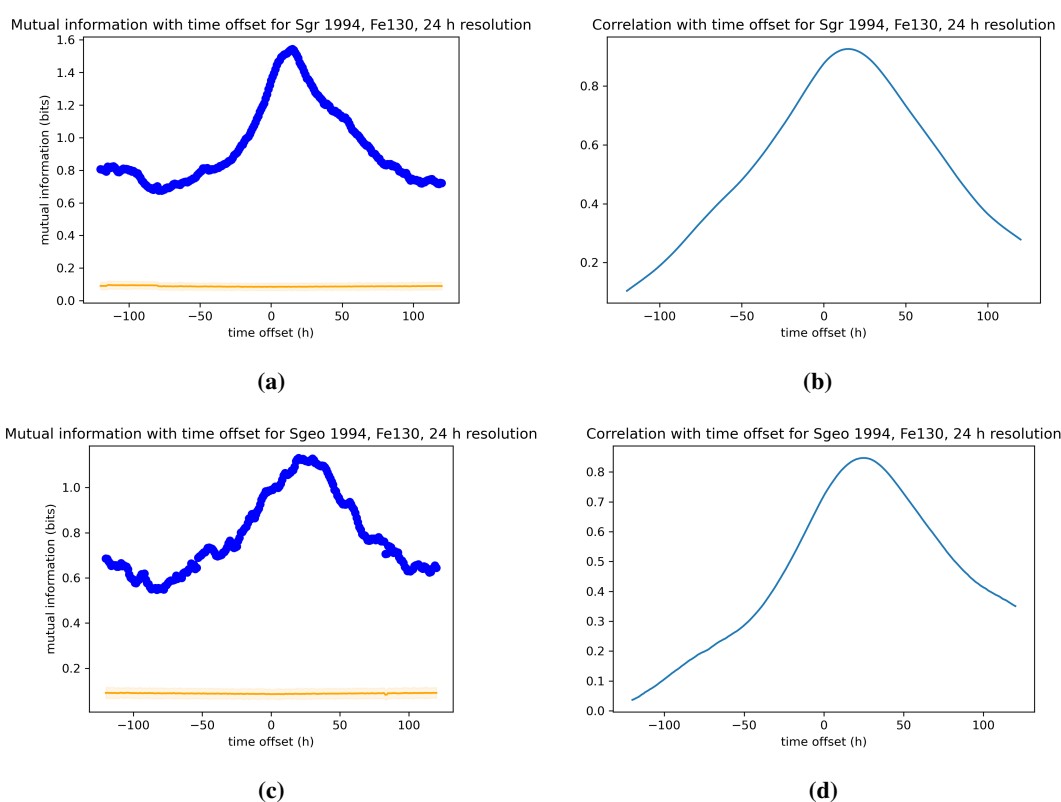

**Figure 9.** Same as in Figure 8 but with a 24 hour moving average of each indices.

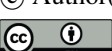

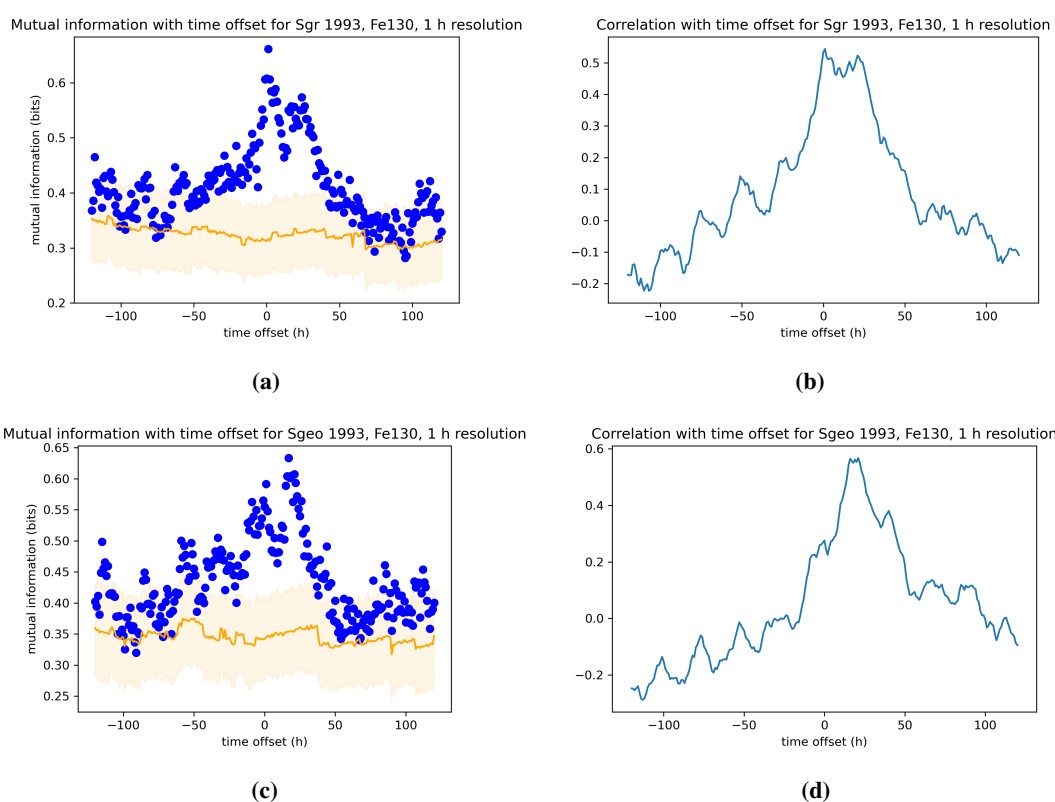

**Figure 10.** Same as in Figure 8 but for the 1993 event of Rostoker et al. (1998).





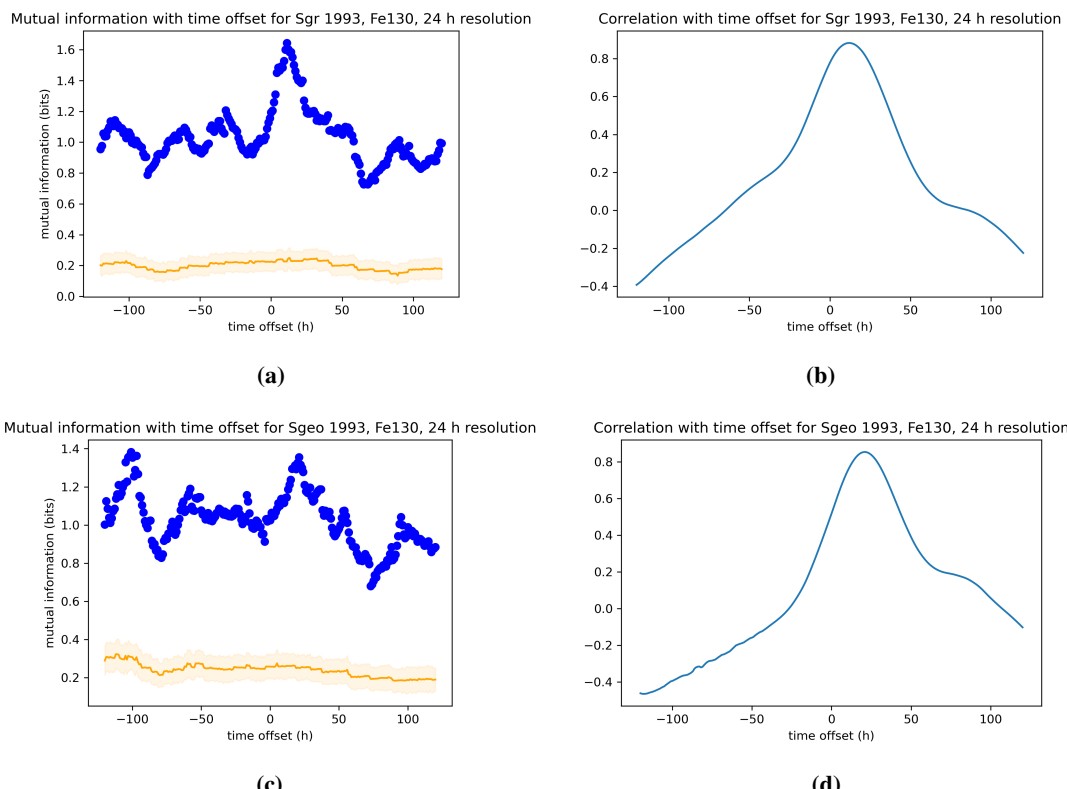

**Figure 11.** Same as in Figure 9 but with a 24 hour moving average.





| Event | Flux index | ULF index | Pearson Correlation | Mutual Information | Adjusted correlation | Lag (hours) |
|---|---|---|---|---|---|---|
| 1993 | Fe130 | Geo | $0.57 \pm 0.03$ | $0.63 \pm 0.04$ | $0.76 \pm 0.02$ | 17 |
| 1993 | Fe130 | Ground | $0.54 \pm 0.03$ | $0.66 \pm 0.04$ | $0.77 \pm 0.01$ | 1 |
| 1994 | Fe130 | Geo | $0.55 \pm 0.01$ | $0.44 \pm 0.02$ | $0.68 \pm 0.02$ | 0 |
| 1994 | Fe130 | Ground | $0.68 \pm 0.01$ | $0.67 \pm 0.01$ | $0.78 \pm 0.01$ | 0 |
| 1993 | Fe1p2 | Geo | $0.41 \pm 0.03$ | $0.42 \pm 0.04$ | $0.66 \pm 0.03$ | 42 |
| 1993 | Fe1p2 | Ground | $0.43 \pm 0.03$ | $0.36 \pm 0.04$ | $0.63 \pm 0.04$ | 50 |
| 1994 | Fe1p2 | Geo | $0.52 \pm 0.02$ | $0.40 \pm 0.02$ | $0.65 \pm 0.01$ | 67 |
| 1994 | Fe1p2 | Ground | $0.59 \pm 0.02$ | $0.49 \pm 0.02$ | $0.70 \pm 0.01$ | 48 |

**Table 1**: Maximum values in correlation & mutual Information for positive lags and associated adjusted correlation.






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
