# Peer review of "Quantifying the nonlinear dependence of energetic electron fluxes in the Earth's radiation belts with radial diffusion drivers"

_Annales Geophysicae, 2021_

## Author Response (AR1)

We thank reviewer 1 for carefully reading the manuscript and providing us with a constructive feedback. Our answer to the specific comments (in bold and italic) can be found below.

Specific Comments:

*1) Section 2 provides~ 100 lines of generalities ("a brief but self-contained tutorial on the Shannon entropy and mutual information for a reader who is not familiar with information" l.84-85). Yet, its value for the remainder of the manuscript is unclear. For instance, Fig. 1 provides the results of a numerical test that appears to be unrelated to the paper.*

Figure 1 is provided to show the correspondence between the Pearson correlation and mutual information. We believe that everyone in our scientific community is familiar with the notion that a Pearson correlation greater than 0.8 is large. But the use of mutual information is fairly recent, and Figure 1 determines what values of mutual information are considered large, i.e., MI> 0.5. This clarification has been added to the revised manuscript and discussion of Figures 4-11 now refer to Figure 1.

*In addition, the formulas used to determine the quantities plotted Fig.4-Fig.11 are not explicitly provided. One way to address this comment could be to shorten the Section 2, retaining only the descriptions of the formulas that are used in the ULF/Eflux data analysis, and referring the interested reader to works already published on this topic (e.g. Wing et al., 2016 and references for information theory lectures). Another way could be to create an appendix for supplementary material.*

We understand. We shortened section 2, added an appendix, and we are now more explicit about the equations used to plot Figures 4-11 in the methodology section.

*2) The main results are presented as a list of 8 different figures (each composed of 4 different panels). Yet, there is little difference between the figures: They all appear to provide similar information (namely, that there exists a significant dependence between waves and particle fluxes). Fig.1 a)-b) is enough to illustrate the approach and the point of the paper is conveyed in Table 1. In that context, everything else could be provided as supporting material. If all 32 panels are really necessary to convey the message of the work, this needs to be explained at the beginning of Section 4.*

We agree with the reviewer that Table 1 contains all the essential results for the paper, and that a reader could simply go through Table 1 and skip the figures. However, Table 1 is extracted from the information found in Figures 4-11, and while the shape of the statistical dependencies are similar, differences between the two Events exist and are noteworthy, such as the larger error for mutual information in the 1993 Event than for the 1994 Event due to a difference in data points. Keeping the figures in the manuscript makes the paper self-contained and allows the reader to see the range of lags for which the mutual information is significantly different from zero. However, the revised manuscript now contains this explanation at the beginning of Section 4 and inform the reader that the main results can immediately be found in Table 1 (See L275-278).
* * *
We thank reviewer 2 for carefully reading the manuscript and providing us with constructive feedback. Our answer to the specific comments (in bold and italic) can be found below.

*1. Line 19, "Their study are..." --> "Their studies are..."*
It is now corrected.

*2. Line 61, the Balikhin et al. citation is missing the publication year*
It is now corrected.

*3. Lines 99-100, it is easy to think of scenarios where one can gain information about the likelihood of event X, given Y. However, it is not so easy to think of scenarios where one can lose information. If X and Y are unrelated then no information is gained about X given Y. Can the authors elaborate on this?*
Indeed, if X and Y are not dependent on one another, we have not lost information. But if a variable X (e.g., ULF wave power) and Y (MeV electron fluxes) are dependent on one another under some conditions (e.g., large solar wind speed ), the removal of the conditions upon which the dependence is strong can result in a loss of information (reduction of mutual information), and thus a loss of knowledge. We have added this clarification to the text with a footnote on page 4 and rewording of the above passage on L104-106.

*4. Lines 168-169, the sentence is a bit awkward. The authors probably want to say ...mutual information and Pearson correlation is an indication that the correlation should not be interpreted linearly (or something like that).*
It is now corrected.

*5. Line 210, there should be a coma between "radial diffusion" and "is a leading".*
It is now corrected.

*6. Line 238, there should be a description that SOPA is an instrument on board of Los Alamos National Laboratory (LANL) spacecraft.*
We have added this description.

*7. Line 256, "... and positive vice-versa". Should this be "...negative vice-versa"?*
Indeed, good catch.

*8. Line 261, should the value above the shaded area represents a mutual information that has least three (not six) sigma significance?*
Yes, it is three sigma. We corrected it.

*9. Lines 260-270 and Figures 4 and 5. One of the main differences between mutual information and correlation in Figures 4 and 5 is that mutual informations consistently have very pronounced secondary peaks at time offset around -100 h*

*whereas the secondary peaks in the Pearson correlations appear to be less pronounced or less significant. Can the authors discuss this?*

This is a good observation. Overall, we show that the Pearson correlation is missing out about 20-30% of the statistical dependence due to its inability to capture nonlinearities. We would need to look at each of these peaks in isolation to quantify this, but we can postulate that the difference we see in secondary peaks has a similar explanation.

*10-   In mutual information plots, Figures 4a, 4c, 5a, and 5c, the secondary peaks probably correspond to negative correlations, as inferred from their Pearson correlation counterparts. The anti-correlations can also be seen in Figures 6 and 7. Can the authors explain this anti-correlation between F1.2 and Sgr and Sgeo at time offset -100 h? The anti-correlations between F130 and Sgr and Sge can also be seen in Figure 11 at about the same time offset.*

In Figure 4, the secondary peak around -100h corresponds indeed to a small negative correlation. This negative correlation can be explained by a loss of relativistic electrons associated to an enhancement in ULF wave power. ULF wave power locally increase the magnitude of the mean magnetic field sampled by the particles. So one possible consequence is that locally cyclotron resonances can take place for particles with higher parallel velocity, and result in enhanced scattering and losses. This is beyond the scope of the paper and it is speculative, but it might be interesting for future study to search for ULF wave modulation of local wave-particle interactions in terms of mutual information.

1.   *Lines 345-346, the authors claim that their results show that quantitatively the dependence is modest. This claim is repeated on line 386 and elsewhere in the manuscript. Table 1 shows adjusted correlations of 0.6 to 0.8.  In many studies of space science, correlations of 0.7-0.8 would be considered strong or very good. "modest" is probably a subjective term.  Can the authors comment on what they would quantitatively consider modest or strong or weak correlations?*

This statement certainly needs clarification. The statistical dependence between 130 keV and ULF wave power is NOT modest since a Pearson correlation of 0.78 and a mutual information of 0.67 is large. However, the Pearson correlation for the relativistic electron and ULF wave power, ranging between 0.4 and 0.59 is modest (significant, but not large) when compared to the statistical dependence between 130 keV and ULF wave power. When compared to previous results, e.g., Simms et al. find correlations ranging between 0.15 and 0.65 between the maximum electron fluxes 48-120 hours after the beginning of a storm and average ULF wave power for a given phase storm. In our case, we correlate the maximum ULF wave power with the maximum electron fluxes measured over an hour interval. We have added this clarification to the revised manuscript and provided more details in our discussion of the results of Simms et al.

---

## Referee Report (RR1)

The study adds significantly to the knowledge of nonlinearities in the radiation belt. The revised manuscript reads well and can be published as is. This reviewer has a three additional minor comments, but it would be up to authors whether they would like to consider them or not in.

1. Lines 56-60, if the authors would like to include an example from solar physics, they may want to cite Wing et al. (2018).

Wing, S., J. Johnson, and A. Vourlidas (2018), Information theoretic approach to discovering causalities in the solar cycle, *Ap J*, **854,** 85, https://doi.org/10.3847/1538-4357/aaa8e7

2. Line 359, according to Merriam Webster dictionary, "cannot" is more commonly used in formal writing than "can not", but both are acceptable English usage.

3. Lines 363-364, introducing solar wind parameters can provide a more complete picture and help with the data interpretation. The statement about having solar wind velocity as a common driver is most likely correct. It likely applies to the dependence on the time scale of 2 days (~50 hr) for MeV electrons. For example, it may explain the positive linear and nonlinear correlation that peaks at time offset around 50 hr in Figures 6 or 7. This may be the time scale for the ULF waves to accelerate electrons to MeV energy range, enhancing the outer radiation belt (Paulikas and Blake, 1979; Reeves et al., 2011). (For 130 keV electrons, the time scale is shorter, e.g., the peaks in Figures 10 and 11 occur at a smaller time offset.) However, for a shorter time scale, in the order of several hours, the common driver may be solar wind density/dynamic pressure. For example, Figures 6c, 6d, 7c, and 7d show negative correlations with broad minima (peaks in the case of mutual information) near 0 hr (or within several hours from 0). An increase in solar wind density can increase solar wind dynamic pressure, which can compress the magnetosphere leading to radiation belt electron loss (magnetopause shadowing). The compression also leads to the growth of the ULF waves, which can redistribute the loss radially.

---

## Author Response (AR2)

*We would like to warmly thank the reviewer for very carefully going through our paper. Her/his suggestions have significantly improved the paper. Our answers to the reviewers' comment can be found below in blue.*

The study adds significantly to the knowledge of nonlinearities in the radiation belt. The
revised manuscript reads well and can be published as is. This reviewer has a three
additional minor comments, but it would be up to authors whether they would like to
consider them or not in.

1. Lines 56-60, if the authors would like to include an example from solar physics, they
may want to cite Wing et al. (2018).

Wing, S., J. Johnson, and A. Vourlidas (2018), Information theoretic approach to
discovering causalities in the solar cycle, Ap J, 854, 85, https://doi.org/10.3847/1538-4357/aaa8e7

*We have now added this reference to Lines 56-60.*

2. Line 359, according to Merriam Webster dictionary, "cannot" is more commonly used
in formal writing than "can not", but both are acceptable English usage.

*We have made that change as well.*

3. Lines 363-364, introducing solar wind parameters can provide a more complete picture
and help with the data interpretation. The statement about having solar wind velocity as a
common driver is most likely correct. It likely applies to the dependence on the time scale
of 2 days (~50 hr) for MeV electrons. For example, it may explain the positive linear and
nonlinear correlation that peaks at time offset around 50 hr in Figures 6 or 7. This may be
the time scale for the ULF waves to accelerate electrons to MeV energy range, enhancing
the outer radiation belt (Paulikas and Blake, 1979; Reeves et al., 2011). (For 130 keV
electrons, the time scale is shorter, e.g., the peaks in Figures 10 and 11 occur at a smaller
time offset.) However, for a shorter time scale, in the order of several hours, the common
driver may be solar wind density/dynamic pressure. For example, Figures 6c, 6d, 7c, and
7d show negative correlations with broad minima (peaks in the case of mutual information)
near 0 hr (or within several hours from 0). An increase in solar wind density can increase

solar wind dynamic pressure, which can compress the magnetosphere leading to radiation
belt electron loss (magnetopause shadowing). The compression also leads to the growth
of the ULF waves, which can redistribute the loss radially.

*This is an important clarification. L364-365 now mention that conditional mutual information for >130 keV would be conditioned on the solar wind density.*